# Unravelling *Chlamydia trachomatis* diversity in Amhara, Ethiopia: MLVA-*ompA* sequencing as a molecular typing tool for trachoma

Anna J. Harte[1]*, Ehsan Ghasemian[1], Harry Pickering[1], Joanna Houghton[1], Ambahun Chernet[2], Eshetu Sata[2], Gizachew Yismaw[3], Taye Zeru[3], Zerihun Tadesse[2], E. Kelly Callahan[4], Scott D. Nash[4], Martin J. Holland[1]

1 The London School of Hygiene and Tropical Medicine, London, United Kingdom, 2 The Carter Center, Addis Ababa, Ethiopia, 3 The Amhara Regional Health Bureau, Bahir Dar, Ethiopia, 4 The Carter Center, Atlanta, Georgia, United States of America

* annaharte27@hotmail.com

## Abstract

Trachoma is the leading infectious cause of blindness worldwide and is now largely confined to around 40 low- and middle-income countries. It is caused by *Chlamydia trachomatis* (Ct), a contagious intracellular bacterium. The World Health Organization recommends mass drug administration (MDA) with azithromycin for treatment and control of ocular Ct infections, alongside improving facial cleanliness and environmental conditions to reduce transmission. To understand the molecular epidemiology of trachoma, especially in the context of MDA and transmission dynamics, the identification of Ct genotypes could be useful. While many studies have used the Ct major outer membrane protein gene (*omp*A) for genotyping, it has limitations.

Our study applies a typing system novel to trachoma, Multiple Loci Variable Number Tandem Repeat Analysis combined with *omp*A (MLVA-*omp*A). Ocular swabs were collected post-MDA from four trachoma-endemic zones in Ethiopia between 2011–2017. DNA from 300 children with high Ct polymerase chain reaction (PCR) loads was typed using MLVA-*omp*A, utilizing 3 variable number tandem repeat (VNTR) loci within the Ct genome.

Results show that MLVA-*omp*A exhibited high discriminatory power (0.981) surpassing the recommended threshold for epidemiological studies. We identified 87 MLVA-*omp*A variants across 26 districts. No significant associations were found between variants and clinical signs or chlamydial load. Notably, overall Ct diversity significantly decreased after additional MDA rounds, with a higher proportion of serovar A post-MDA.

Despite challenges in sequencing one VNTR locus (CT1299), MLVA-*omp*A demonstrated cost-effectiveness and efficiency relative to whole genome sequencing, providing valuable information for trachoma control programs on local epidemiology. The findings suggest the potential of MLVA-*omp*A as a reliable tool for typing ocular Ct and understanding transmission dynamics, aiding in the development of targeted interventions for trachoma control.

**Data Availability Statement:** OmpA sanger sequences have been uploaded to genbank via bankit, with accession numbers PP259191-

PP259335. WGS sequences are available via EBI (PRJEB38668).

**Funding:** This work received financial support from the Coalition for Operational Research on Neglected Tropical Diseases, which is funded at the Task Force for Global Health by the Bill & Melinda Gates Foundation, the United Kingdom Department for International Development, and the United States Agency for International Development. Additional financial support was received from the International Trachoma Initiative. Ehsan Ghasemian's salary is funded by the Austrian Science Fund (FWF) [Project Number J- 4608]. The funders had no role in study design, data collection and analysis, decision to publish, or preparation of the manuscript.

**Competing interests:** The authors have declared that no competing interests exist.

## Author summary

Trachoma is the leading infectious cause of blindness worldwide and is largely confined to low- and middle-income countries. It is caused by *Chlamydia trachomatis* (Ct), a contagious intracellular bacterium. The World Health Organization recommends mass drug administration (MDA) with the antibiotic azithromycin for treatment of ocular Ct infections, alongside improving facial cleanliness and environmental conditions to reduce transmission. In most regions MDA is successfully reducing trachoma prevalence to the point where it is no longer a public health issue, however in some areas trachoma persists despite multiple years of interventions. To investigate why trachoma persists, especially in the context of MDA and transmission dynamics, the identification of Ct sequence types may aid in understanding and gauge progress of trachoma control. Our study applies a Ct typing system new to trachoma, which augments the standard method by adding three loci with high mutation rates. Results show that the typing system was able to discriminate between variants with greater resolution than the standard method, and was both cost-effective and more efficient relative to the gold-standard of whole genome sequencing. The findings suggest that this novel method is a reliable tool for typing ocular Ct, which can aid in the development of targeted interventions for trachoma control through improved understanding of Ct transmission.

## Introduction

Trachoma is the leading infectious cause of blindness worldwide [1] and is found primarily in about 40 low- and middle-income countries (LMIC). It is caused by *Chlamydia trachomatis* (Ct), a contagious intracellular bacterium with 4 serotypes (A, B, Ba and C) that are typically responsible for ocular infection in trachoma endemic countries. The World Health Organization (WHO) recommends the use of the Surgery, Antibiotics, Facial Cleanliness, and Environmental improvement (SAFE) strategy, including mass drug administration (MDA) of the antibiotic azithromycin for controlling trachoma [1]. Multiple rounds of MDA are usually required which is determined by the prevalence of clinical signs within the district. Reinfection can occur within months of treatment [2–4], and the clinical signs of disease often persist in the population while ocular Ct infection is low or absent [5,6]. The surveillance of Ct strains is of interest for monitoring transmission dynamics, as well as identifying the potential dominance of individual strains, which may indicate selection by antibiotics or increased virulence, persistent infection and/or re-infection [7].

 Molecular epidemiological studies of Ct infection within communities often focus on the outer membrane protein A (*omp*A) gene, as this codes for the polymorphic major outer membrane protein (MOMP), a key chlamydial antigen targeted by the host immune system and one therefore theoretically subject to immune selection [8–11]. However, previous work on genital (D-K) and Lymphogranuloma venereum (LGV) serovars has demonstrated that *omp*A genotyping alone has insufficient discriminatory power [12–15] for the reliable identification of different variants within a population [16]. Since recombination within *omp*A has been demonstrated, this may also obscure phylogenetic relationships [17]. Whole-Genome Sequencing (WGS) should offer the highest resolution for variant identification; however, it remains prohibitive in terms of cost, equipment, high technical expertise and large sample load required, placing this beyond the access and feasibility for trachoma control programmes operating in LMIC. To address this, several polymerase chain reaction (PCR)-amplicon sequence-based typing methods have been developed, where two or more loci are amplified

and then subjected to chain termination sequencing (Sanger sequencing) for each individual sample.

These typing systems have been used to investigate Ct strain diversity, each with varying levels of discriminatory power [7]. Multi-locus sequence typing (MLST) and multiple loci variable number of tandem repeat analysis (MLVA) are two types of PCR sequencing that use sequence types (ST) to define different variants. Classic MLST is focused on sequence variation in loci that are not considered to be under selection pressure and are stable housekeeping genes [18,19] and is best suited to long-term and global epidemiology. Other MLST studies have used highly variable loci instead, to improve variant resolution on a smaller epidemiological scale [20–22]. MLVA provides a further alternative and uses variable number tandem repeats (VNTR), which have a higher rate of mutations [23] and can therefore provide a useful identifier of novel strains for small scale, local epidemiological studies [22]. MLST methods have been applied to ocular serovars and in trachoma epidemiology [20,24,25], however, MLVA has not yet been used to investigate the molecular epidemiology of trachoma.

In this study conjunctival DNA collected from children living in trachoma-endemic areas of Amhara, Ethiopia, which were surveyed between 2011 and 2017, were used to sequence three VNTR loci and *omp*A as outlined by Pedersen et al., [26]. The aim of this study was to determine the efficacy of MLVA-*omp*A when applied to ocular samples and test the discriminatory power (DP). We then investigated whether MLVA-*omp*A variants were associated with clinical signs of trachoma and the impact of MDA on variants in the population. We propose that MLVA-*omp*A typing could serve as a tool for the surveillance of emerging Ct variants within the context of localized epidemiological studies. Additionally, it can be employed to establish a profile of the evolutionary dynamics of Ct variants over time within populations remaining endemic to trachoma.

## Methods

### Ethics statement

The ethical approvals associated with the collection and processing of these samples have been described previously [27–29]. Survey methods were reviewed and approved by the Emory University Institutional Review Board (protocol 079–2006) as well as by the Amhara Regional Health Bureau. Due to elevated levels of illiteracy within the community, permission was granted for obtaining verbal consent or assent. Electronic recording of oral consent or assent was implemented for all individuals, aligning with the principles outlined in the Declaration of Helsinki. Participants had the option to conclude the examination at any juncture without the need for providing an explanation.

### Study population

The SAFE strategy was scaled up between 2007 and 2010 to reach all districts in Amhara. This included community-wide MDA with antibiotics, extensive health campaigns focusing on proper hygiene and face washing, and advocacy for the construction and use of latrines [29–32] Between 2011 and 2015 all districts were then surveyed to measure the impact of approximately 5 years of SAFE interventions on trachoma prevalence [29,33]. Impact surveys use multi-stage sampling, and trained and certified graders use the WHO-simplified grading system [34] to determine the presence of the trachoma clinical signs trachomatous inflammation-follicular (TF), trachomatous inflammation-intense (TI) and trachomatous scarring (TS), and to estimate the prevalence of water, sanitation and hygiene (WASH) indicators such as facial cleanliness and improved water and latrine presence [29]. As part of these surveys, conjunctival swabs were collected to estimate population-based prevalence of Ct infection among

children aged 1 to 5 years [33]. Impact surveys, including conjunctival swabbing, were repeated throughout the region between 2014 and 2021 after an additional 3 to 5 years of interventions [28,30]. The samples used in this study were a subset of those collected between 2011–1015 from 58 surveyed districts which make up four zones (North Gondar, South Gondar, East Gojam and Waghemra) [28]. An additional subset of samples used in this study came from all 11 districts in South Gondar zone, collected between 2014 and 2017 as part of the second round of surveys. Each individual was given a unique ID code which was not known to anyone outside of the research group [27–29].

## DNA extraction and sample preparation for sequencing

The initial testing of samples was performed at the Amhara Public Health Institute in Bahir Dar, Ethiopia. Swabs were randomised and pooled into batches of five samples per pool, and tested for two highly conserved Ct plasmid targets using the Abbott Realtime assay on the Abbott m2000 (Abbott Molecular Inc., Des Plaines, IL, USA) [33]. For the subset of districts included in this study, individual samples from positive pools were assayed again to determine individual level infection. Ct load was determined by converting the Abbott m2000 delta cycle to elementary body (EB) count using an EB standard curve, generated using a standard set of EB titrations [28,35]. There were 525 Ct positive samples collected from these study districts. 300 samples with the highest Ct load were chosen for further sequencing analysis. DNA was re-extracted from all 300 conjunctival swabs as previously described [27]. Ninety-nine samples with a sufficiently high Ct load were chosen for WGS, the results of which have already been published, including the bioinformatic extraction of the *ompA* region [27]. For the remaining 201 samples, *ompA* was sequenced by Sanger sequencing, however for the three VNTRs of interest, CT1291, CT1299 and CT1335, all 300 samples were sequenced by Sanger sequencing Reference sequences from serovars A (A/2497: Acc. FM872306; A/HAR13: NC_007429), B (B/Jali20/OT: Acc. FM872308; B/Tunis864: ERR12253485), C (C/TW-3, Acc. NC_023060), D (D/UW3/CX, Acc. NC_000117), and L (L3/404/LN: Acc. HE601955; L2/434/Bu: Acc. AM884176) were also sequenced for the 3 VNTR regions to compare between published WGS sequences.

## *ompA* Sanger sequencing preparation

Prior to sequencing, the extracted eye swab DNA underwent one or two rounds of PCR using a nested PCR, following the procedure outlined in Andreasen et al. 2008, [11]. For the first round, 20 µL of 5prime MasterMix (QuantaBio), 20 µL of molecular grade water (Corning, Manassas, VA20109 USA), 2.5 µL each of forward and reverse primers *ompA*-87 and *ompA*-1163 and 5 µL of DNA template were run using the cycling conditions as follows: Initial denaturation of 2 min at 94˚C, then 35 cycles of 94˚C for 15 S and 62˚C for 75 S, and a final elongation step of 72˚C for 10 min. This PCR resulted in a product of 1076 base pairs (bp) in size. Samples were run on a gel, and if no band was present, nested PCR was performed. For the second round of PCR, 10 µL 5Prime HotMasterMix (QuantaBio), 1.25 µL *ompA*-87 and 1.25 µL *ompA*-1059 primers and 11.5 µL molecular grade water (Corning, Manassas, VA20109 USA) were added to 1 µL of a 1 in 200 dilution of the PCR product from the first round. The final product size of the target sequence (*ompA*-inner) was 972 bp in length. This product was then cleaned using a ratio of 0.8 X final volume of AMPure XP magnetic beads (Beckman Coulter) following the manufacturer's instructions, quantified on the Qubit 2.0 Fluorometer (Thermo Fisher) and diluted to a concentration of ~10 ng/µL. Samples were sent to Source BioScience (Cambridge, UK) with *ompA*-97 forward primer and/or *ompA*-1059 for samples sequenced in both directions.

## VNTR sequencing preparation

All 300 samples were processed for the VNTRs, with a separate PCR reaction used to amplify each VNTR region. DNA extracted from 8 Ct reference samples were also sequenced for comparison with existing genomic data. Samples were run in 25 µL reaction volumes, using 5 µL DNA template, 10 µL Accustart (Quantabio, Hilden, Germany), 1.25 µL each forward and reverse primers, and 7.5 µL molecular grade water (Corning, Manassas, VA20109 USA). Three different sources of primers were used, due to complications with the proximity of some of the VNTR regions to the start of the sequencing, and the subsequent low-quality bases associated with the first 30–50 bps. For primers sourced from Pedersen et al., [26] and the new CT1299 forward primer, cycling conditions were 10 min held at 94˚C, then 40 cycles of 45 S at 94˚C, 20 S at 59˚C, 20 S at 72˚C, hold at 4˚C. For primers sourced from Labiran et al., [36] the annealing temperature was 56˚C for CT1335 and 60˚C for CT1299. Samples were sent to Source BioScience (Cambridge, UK) with 3.2 pmol of forward primer for each VNTR. CT1299 was sequenced in both the forward and reverse direction for some samples with low quality sequencing due to extended repeat base regions. Primers and amplicon sizes for the 3 VNTR regions are listed in Table 1. Samples that did not produce sufficient sequencing quality to call the sequence type in the first round of sequencing for CT1299 were re-run using a modified CT1299 forward primer to increase the distance between the start of the sequence and the flanking and VNTR region [37]. A total of 140 samples were tested using the new CT1299 forward primer.

## VNTR calling and MLVA-*omp*A

Only VNTR sequences produced by Sanger sequencing were used in further analysis. Sequences were aligned within each respective VNTR using MEGA-X (Version 11.0.13), and the sequence type was recorded by manually counting the number of repeat base pairs as well as the respective flanking regions, following the typing method outlined by Pedersen et al., [26] and Wang et al., [38]. Chromatograms were checked to ensure that base-calling was accurate, and sequences were discarded from further analysis if a specific VNTR type could not be assigned. References where WGS sequences were available on National Center for Biotechnology Information (NCBI) from serovars A (A/2497; A/HAR13), B (B/Jali20/OT; B/Tunis864), C (C/TW-3), D (D/UW3/CX), and L (L3/404/LN; L2/434/Bu) were used to compare VNTR-amplicon Sanger sequencing against WGS derived VNTR sequence, alongside comparison of published WGS data from a subset of samples from this study [27] by calculating the kappa statistic.

**Table 1. Forward and reverse primers used to amplify VNTR regions and *omp*A of *Chlamydia trachomatis* in Ethiopian conjunctival samples collected between 2011–2017, for Sanger sequencing and the amplicon size produced.** VNTR: Variable number tandem repeat.

| Target | Source | Forward primer | Reverse primer | Amplicon size |
|---|---|---|---|---|
| *omp*A | Andreasen et al., [11] | TGAACCAAGCCTTATGATCGACGG-(*omp*A-87) | CGGAATTGTGCATTTACGTGAG-(*omp*A-1163) | 1076 |
| *omp*A-inner | Andreasen et al., [11] | TGAACCAAGCCTTATGATCGACGG-(*omp*A-87) | GCAAGATTTTCTAGATTTCATC-(*omp*A-1059) | 972 |
| CT1291 | Pedersen et al., [26] | GCCAAGAAAAACATGCTGGT | AGGATATTTCCCTCAGTTATTCG | 225 |
| CT1299 | Labiran et al., [36] | ATCGCTTAAGATTCTCGGAGG | AGGTTCTAGCTGAGCATGGG | 342 |
| CT1299 (new) | This study and Labiran et al., [36] | GGAATTTCCATAGACGGTTGATA | AGGTTCTAGCTGAGCATGGG | 381 |
| CT1335 | Labiran et al., [36] | AAAGCGTCCTCTGGAAGGG | CCTTCTCCTAACAACTTACGC | 208 |

### *omp*A serovar calling

*omp*A sequences were extracted from WGS data generated from the 99 samples originally ana-lysed in Pickering et al., [27]. All WGS sequences were aligned to *omp*A reference sequences A/HAR13, B/Jali20 and C/TW-3 using bowtie2 [39], and samtools [40] was used to extract the matching *omp*A sequences. The best alignments were selected based on the minimum number of Ns present in the sequence. The 146 Sanger-derived *omp*A sequences, were separated based on serovar, and aligned within each serovar group to a reference sequence using MEGA-X; Serovar A (A/HAR13, Acc. DQ064279), serovar B (B/TW-5/OT, Acc. M17342) and serovar Ba (VR-347-Ba, Acc. KP120856).

### Variant types and typeability

Individual variant types were classified following Pedersen et al., [26], Wang et al., [38], and Manning et al., [15] including both the VNTR and the associated flanking region. The combi-nation of all three VNTRs within a sample was used to assign MLVA type, with the addition of each *omp*A type to create the MLVA-*omp*A variant. Variants are technically the same as strains, however a variant is only referred to as a strain when it shows distinct physical proper-ties, such as the reference strains used for alignments. As we have not investigated any poten-tial physical properties of the variants identified in this study only variant has been used. Typeability was calculated by dividing the number of samples that were successfully sequenced for all three VNTRs and *omp*A (MLVA-*omp*A) by the total number of samples available (n = 300).

### Diversity and discriminatory power

DP is defined as the ability to distinguish different variants, and is expressed as the probability that two different variants will be placed into different categories. The level of diversity was cal-culated for each genotyping system with and without associated VNTR-flanking regions (indi-vidual VNTRs, MLVA, *omp*A, MLVA- *omp*A) using Simpson's index of diversity following the recommended modification outlined by Hunter and Gaston [41]. Hutcheson's *t*-test was used to compare the Simpson diversity indices for pre- and post-MDA time points for the South Gondar zone. Data was available from two different time points for South Gondar because surveys were performed after 5 rounds of MDA and 8–10 rounds of MDA. Zonal diversity was compared between the two time points using Hutcheson's *t*-test in the vegan package [42] in R, to determine if there was a significant change in the level of diversity after extra rounds of MDA.

### Association analyses

Association analyses were performed to investigate the relationship between individual level TF and/or TI and the MLVA-*omp*A variants. Due to the high number of low frequency vari-ants, variants were stratified as; high (>4), medium [2–4] and low [1] frequency, and treated as a categorical variable, with the high frequency group set as the reference. Mixed effects logis-tic regression (package lme4, R, [43]) was performed for each clinical sign, with the 3 variant categories as the independent variable, adjusting for age, gender and district as a random effect.

Association analyses were also performed using village-level TI and TF as the independent variable using generalised linear mixed effects models (Package glmm, R) [44], adjusting for age, gender and district as a random effect. The same model was run using Ct load as the inde-pendent variable. District-level Ct, TI and TF prevalence were also used as dependent variables

to test the relationship with district diversity, accounting for the number of samples per district.

## Minimum spanning tree and phylogenetic tree analysis

Minimum spanning trees (MSTs) were created using Phyloviz 2.0 software [45]. Variants based on the Pedersen et al., [26] typing system, updated to include the flanking regions by Wang et al., [38], were used to create MSTs of all available variants, overlaid with data on serovar, year of collection, and zone. For the tanglegram construction, we developed two phylogenetic trees. The first tree was constructed using the 3 VNTR sequences and their flanking regions obtained from Sanger sequencing, while the second tree utilized sequences obtained from WGS. Sequences for each sample were concatenated using Geneious (Version 2023.2.1), and alignments were performed using MAFFT (Version v7.490) with a 200 PAM/K = 2 scoring matrix [46,47]. The Neighbor-Joining method, incorporating the Jukes-Cantor genetic distance model with 100 bootstraps and no outgroup, was employed to construct the trees using Geneious Tree Builder. Subsequently, the trees were exported to Dendroscope (version 3.8.10) for tanglegram visualization [48].

## Results

### *omp*A and VNTR Sequences

*OmpA* serovar type derived from WGS was obtained for all 99 Ct WGS. A further 201 samples that were sequenced for *ompA* using Sanger sequencing generated 146 successful sequences (72.6%). The 55 unsuccessful sequences were due to unreadable chromatograms containing a large proportion of Ns, due to either poor quality DNA or overlapping peaks. These could potentially represent mixed infection with multiple variants however these results cannot be reliably resolved and were excluded. Data was missing in relatively equal proportion over the six years that samples were collected (S1 Table). For the 3 VNTRs generated by Sanger sequencing, CT1291 was successfully sequenced in 266/300 samples (88.6%), CT1299 in 252/300 (84%) and CT1335 in 255/300 (85%). In total, 194/300 (65%) samples had sequencing results of a high enough quality to call serovar and VNTR sequence types for all 3 VNTRs (Fig 1). Sequencing of *ompA* and the 3 VNTRs produced variable results, with CT1291 the most successful in terms of typeability (88.7%) and CT1299 the least (84.0%) (Table 2, Fig 2). There were consistent difficulties with sequencing the repeat region of CT1299, with samples exceeding 14Cs demonstrating evidence of polymerase slippage and PCR stutter [23]. Samples with greater than 14Cs in a row were therefore sequenced in both the forward and reverse directions, however, 10 CT1299 samples with 15–18 repeat Cs were not able to be confidently identified, and were removed from the dataset.

### VNTR and MLVA-*omp*A typeability

The typeability of each VNTR and *omp*A individually were above 80%, however due to a small proportion of dissimilarity between the samples that were not successful for each round of sequencing (Fig 2), the overall typeability of MLVA-*omp*A for this study was 64.6%. There was a mix of serovars A, B and Ba across the entire Amhara region from samples collected between 2011–2015 (Fig 3). A total of 87 variants were found within this population (DP = 0.981). WGS sequences from reference serovar STs deposited in NCBI matched our sample sequencing results for six out of eight reference genomes, with the repeat C region in CT1299 resulting in a low quality read for C/TW-3 and a mismatch in CT1299 region for B/Tunis864 (Table 3).

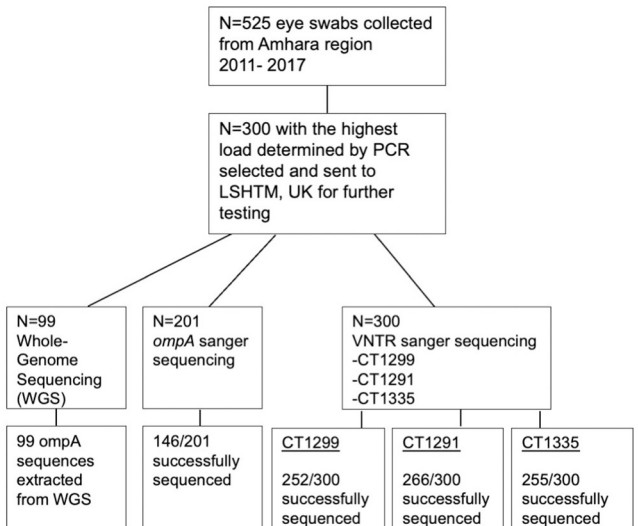

**Fig 1. Workflow diagram showing the process of conjunctival swab samples collected between 2011–2017 from the Amhara region of Ethiopia, and the proportion successfully sequenced for each VNTR and *ompA*.** LSHTM: London School of Hygiene and Tropical Medicine. VNTR: Variable number tandem repeat.

CT1299 and CT1291 had the highest number of STs with 12 and 8 respectively, with CT1335 having 6 STs. Three new STs were recorded for CT1291, two new STs for CT1299 and four new STs recorded in CT1335 (Table 4). The frequency of the STs for each VNTR are listed in Table 4a–4c. Frequencies for both MLVA and MLVA-*omp*A show a relatively high proportion of variant types are singletons since almost half of the MLVA-*omp*A variant types have a frequency of one.

## Minimum spanning trees

The MLVA-*omp*A MST (Fig 4) showed limited defined clustering when separated by year of collection. There was a small cluster from 2013, however the most common variants, represented by larger circles, were present in multiple years. There were also some defined clusters by *omp*A serovar (S1 Fig), however the MSTs categorised by rounds of MDA (S2 Fig) and zone (S3 Fig) showed limited clusters of variants. There was a small (n = 10) defined cluster of variants collected in East Gojam in 2013 that were all *omp*A serovar B/Ba.

**Table 2. Typeability and discriminatory power for each VNTR separately, the VNTRs collectively (MLVA), *ompA* and MLVA-*ompA*.** Discriminatory power was determined following Hunter and Gaston's modified Simpson's diversity index (Hunter and Gaston, 1987). Typeability is calculated by the number of successful sequencing reactions divided by the total number of samples available. The number of variants for the *ompA* sequencing refers to *ompA* serovars, as determined by homology of each sequence within the BLAST-n NCBI database.

| | CT1291 | CT1299 | CT1335 | MLVA | *ompA*-genovar | *ompA*- variant type | MLVA-*ompA* |
|---|---|---|---|---|---|---|---|
| Total (300) | 266 | 252 | 255 | 232 | 245 | 240 | 194 |
| Typeability (%) | 88.7 | 84.0 | 85.0 | 77.3 | 81.7 | 80.0 | 64.6 |
| Number of sequence types | 8 | 12 | 6 | 66 | 3 | 17 | 87 |
| Discriminatory power | 0.676 | 0.812 | 0.530 | 0.959 | 0.504 | 0.820 | 0.981 |

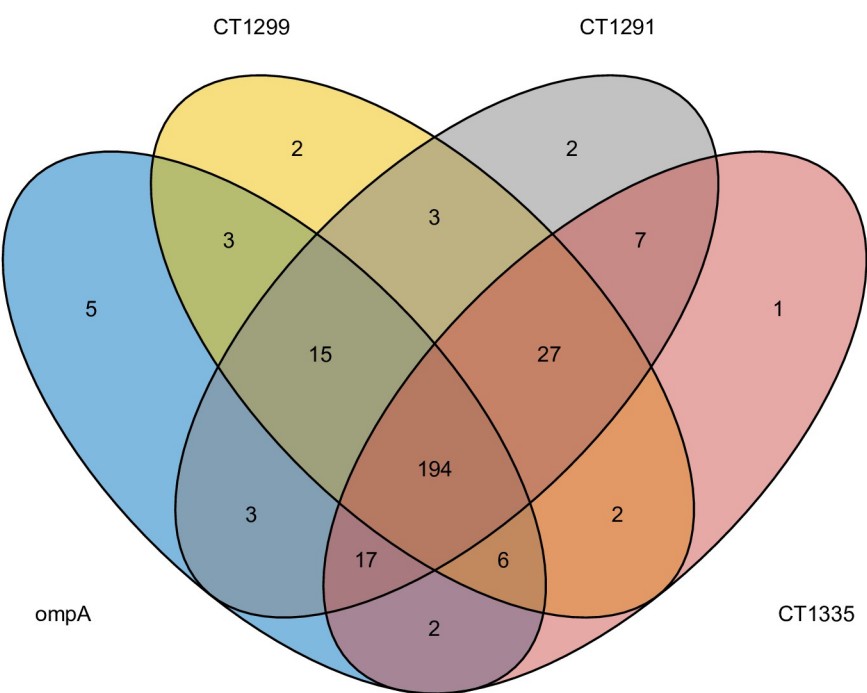

**Fig 2. Venn diagram illustrating the overlap in successful Sanger sequencing of each conjunctival swab sample collected from Amhara, Ethiopia between 2011–2017 for the 3 VNTRs and *ompA*.** Nine samples were not able to be sequenced for any of the VNTRs or *ompA*. VNTR: Variable number tandem repeat.

## Phylogenetic trees

A total of 45 samples provided complete VNTR sequences along with their flanking regions through WGS, which were subsequently utilized to construct a Neighbor-Joining tree (Fig 5). The remaining 44 samples that were unable to provide a complete VNTR sequence were missing loci due to insufficient sequencing quality in either WGS or sanger sequences or both (S2 Table). Simultaneously, the sequences obtained from Sanger sequencing for these samples were employed for a parallel Neighbor-Joining tree construction (Fig 5). Through the tanglegram, a comparative analysis between the trees generated from Sanger sequencing and WGS data revealed a more intricate structure in the Sanger data. In this context, Ethiopian sequences corresponding to each variant clustered together, forming distinct branches separate from other variants. Conversely, the WGS derived sequences from Ethiopia exhibited a lack of differentiation based on assigned STs. Notably, 39 out of the 45 sequences grouped closely with the reference strain A/HAR13, suggesting a higher degree of homogeneity within this subset.

## WGS derived and PCR amplicon VNTR sequence comparisons

The 99 samples that underwent WGS had fair consistency with Sanger sequencing results for CT1335 (weighted kappa = 0.21), however poor consistency for CT1291 (weighted kappa = 0.16). Only 3 CT1335 sequences were unable to be assigned using the WGS sequences, which coincided with 3 samples where the VNTR type was 9T/8A, as opposed to the most frequent 10T/8A. There was a discrepancy between the CT1335 flanking region between WGS and Sanger sequences, with Sanger sequences identifying some samples carrying the ST with a

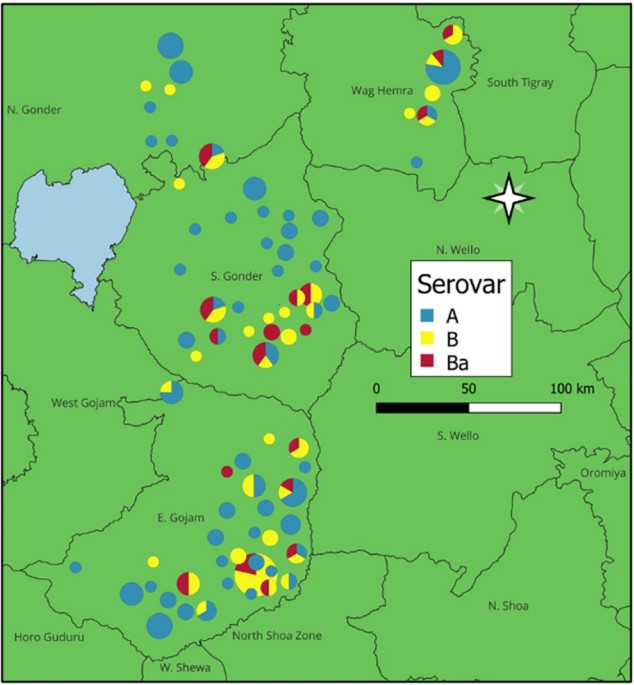

**Fig 3. Map of the Amhara region of Ethiopia showing the location and proportion of *Chlamydia trachomatis* serovars A, B and Ba at the first sampling time points (2011–2015) from conjunctival eye swab samples collected after the first five rounds of mass drug administration.** The size of the pie chart is proportional to how many samples are represented, with larger pie charts indicating more samples. Maps were sourced from the Trachoma Atlas (https://atlas.trachomadata.org/) which uses Open Street Maps (Mapbox).

flanking sequence of four repeated Adenosine bases (**GAAAA**GG) whereas WGS identified them as having five (**GAAAAA**GG) (S2 Table).

CT1291 sequences were unable to be assigned in 19 of the WGS sequences (19%), and were inconsistent with the VNTR type called by Sanger sequencing in 45 individuals (45%). For the CT1291 flanking region nine samples were unable to be called (9%), with three mismatches between Sanger and WGS.

**Table 3.** ***Chlamydia trachomatis*** **(Ct) reference strain sequencing results for each VNTR by Sanger sequencing ("this study") in comparison to VNTRs extracted from whole-genome sequences sourced from NCBI ("reference").** Reference strain DNA was extracted from cells inoculated with bacteria prior to the commencement of this study.

| Serovar ID | CT1299 | | CT1291 | | CT1335 | | Reference Accession number |
|---|---|---|---|---|---|---|---|
| | This study | Reference | This study | Reference | This study | Reference | |
| L2/434/Bu | 6C | 6C | 9C | 9C | 11T/8A | 11T/8A | AM884176 |
| B/Jali20 | 8C | 8C | 10C | 10C | 10T/8A | 10T/8A | FM872308 |
| B/Tunis864 | 13C | 12C | 7C | 7C | 10T/8A | 10T/8A | ERR12253485 |
| D/UW3/CX | 12C | 12C | 10C | 10C | 10T/8A | 10T/8A | NC_000117 |
| C/TW-3 | >17C | 14C | 9C | 9C | 10T/8A | 10T/8A | NC_023060 |
| A/HAR13 | 14C | 14C | 9C | 9C | 10T/8A | 10T/8A | NC_007429 |
| A/2497 | 12C | 12C | 8C | 8C | 10T/8A | 10T/8A | FM872306 |
| L3/404/LN | 6C | 6C | 7C | 7C | 11T/8A | 11T/8A | HE601955 |

**Table 4. VNTR sequence types (ST) and frequency table with Pedersen et al., [26] and Wang al., [38] type codes for CT1335, CT1299 and CT1291. Flanking regions forward and after the repeat nucleotide regions are included.** Only sequence types found in this study are shown. Novel STs are identified with an asterisk *.

| VNTR | Pedersen VNTR STs | Flanking region and repeat sequence | Frequency |
|---|---|---|---|
| CT1335 | 1a* | GAAAAGG-**9T/8A**-GCTTTTGT | 5 |
| | 2 | GAAAA̲AGG-**10T/7A**-GCTTTTGT | 17 |
| | 2a* | GAAAAGG-**10T/7A**-GCTTTTGT | 11 |
| | 3 | GAAAA̲AGG-**10T/8A**-GCTTTTGT | 35 |
| | 3a* | GAAAAGG-**10T/8A**-GCTTTTGT | 125 |
| | 14* | GAAAAGG-**9T/7A**-GCTTTTGT | 1 |
| CT1299 | 2 | TTATTCT-8C-ATCAAA | 1 |
| | 3 | TTATTCT- 9C-ATCAAA | 19 |
| | 4 | TTATTCT- 10C-ATCAAA | 64 |
| | 4b* | TTATTCT- 10C- T5C-ATCAAA | 1 |
| | 5 | TTATTCT-11C-ATCAAA | 36 |
| | 6 | TTATTCT- 12C-ATCAAA | 11 |
| | 7 | TTATTCT- 13C-ATCAAA | 21 |
| | 8 | TTATTCT- 14C-ATCAAA | 40 |
| | 8b* | TTATTCT- 14C-T5C-ATCAAA | 1 |
| CT1291 | 2 | AAAATGGTCT-8C-TATTG | 28 |
| | 2c* | AAAATGGT-8C-TATTG | 14 |
| | 3 | AAAATGGTCT-9C-TATTG | 99 |
| | 3c* | AAAATGGT-9C-TATTG | 3 |
| | 4 | AAAATGGTCT-10C-TATTG | 38 |
| | 5 | AAAATGGTCT-11C-TATTG | 7 |
| | 7 | AAAATGGTCT-12C-TATTG | 1 |
| | 8* | AAAATGGTCT-7C-TATTG | 4 |

CT1299 VNTR sequences were unable to be called in 35 of the WGS sequences (35%), and in the remaining 64 sequences, 61 were inconsistent with the VNTR type called by Sanger sequencing (95%). The CT1299 flanking region did not show any variation in STs in Sanger sequences, however in WGS sequences there were 3 sequences that had the flanking sequence "ATTCT" repeated twice, which was not present in the Sanger sequence of the same sample (S2 Table).

## Association analyses

In the dataset available for analysis, 159 people had TF, 85 had TI, and 2 had TS. A total of 176/194 people (91%) had active trachoma (TF and/or TI). There were no significant differences between the variant frequency groupings or serovar and either TI or TF clinical signs, for both absence/presence analyses, village-level prevalence or Ct load (Table 5). There was a significant difference in MLVA-*omp*A diversity between 5 years of MDA and 8–10 years of MDA in South Gondar ($p = 0.006$), with less diversity after extra rounds of MDA (Fig 6). There was an observable *ompA* serovar switch over time (Fig 7), with a higher proportion of serovar A (91%) after the additional rounds of MDA (46%) (Fig 6).

## Discussion

Understanding Ct transmission dynamics is an important part of trachoma control, as the identification of Ct genotypes can provide information on the infection source, the presence of

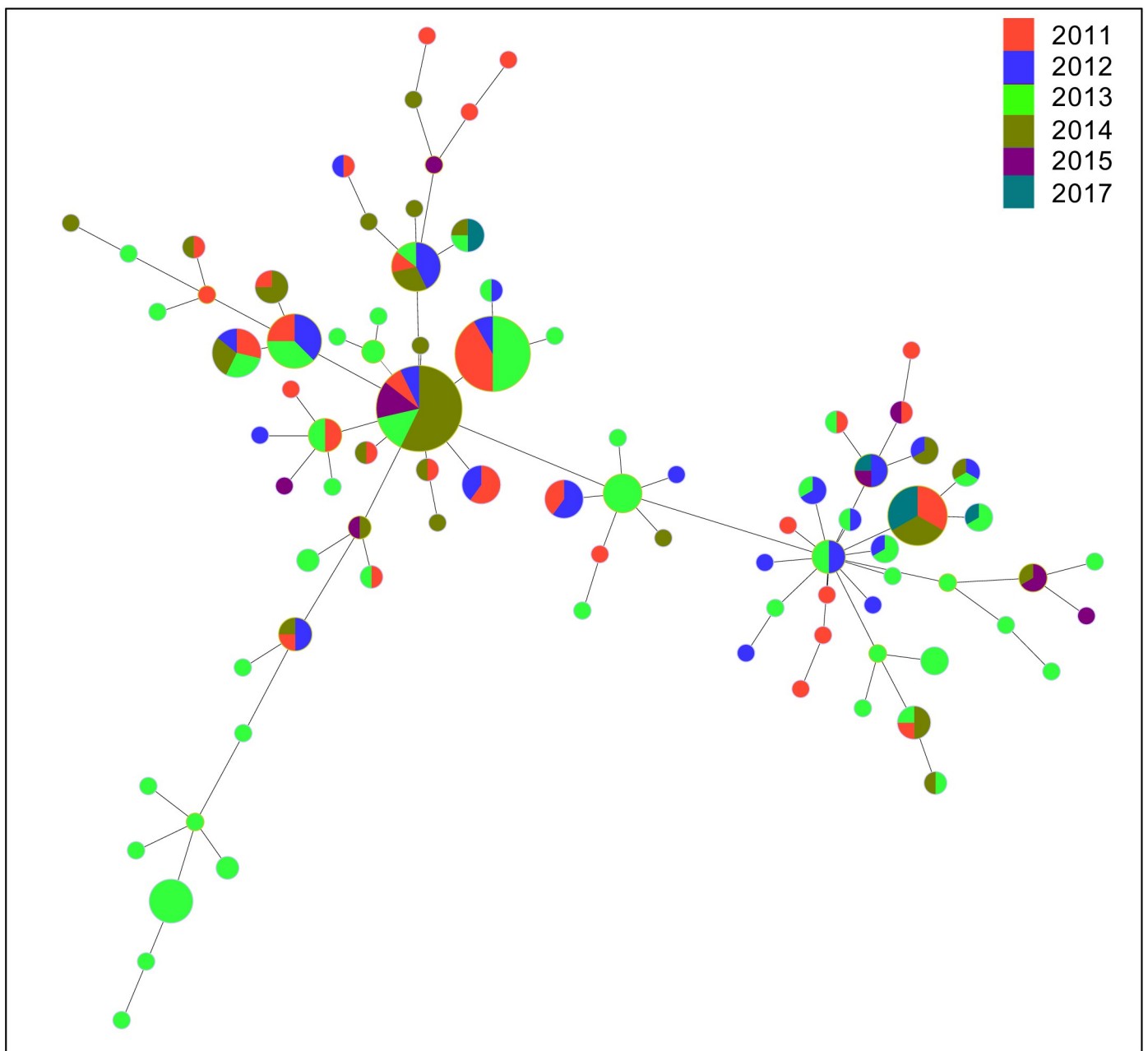

**Fig 4. A minimum spanning tree showing the sequence-types identified in the Amhara region of Ethiopia between 2011–2017.** Each MLVA-*omp*A sequence-type (n = 87) is coloured by year of collection. Each circle represents a different sequence-type, with the size of circle being directly proportional to the number of individuals who had that variant. Trees were generated using Phyloviz 2.0.

repeat or persistent infections, and the impact of antibiotic treatment and other interventions [49]. Different ocular genotypes of Ct may also elicit varying degrees of infection intensity and disease severity [50]. The majority of epidemiological studies have primarily focused on *omp*A, however, this has been shown to be limited in terms of discriminatory power [7]. The results from this study have demonstrated that the use of MLVA-*omp*A can provide a sufficient resolution for molecular epidemiology studies in trachoma since it exceeds the recommended

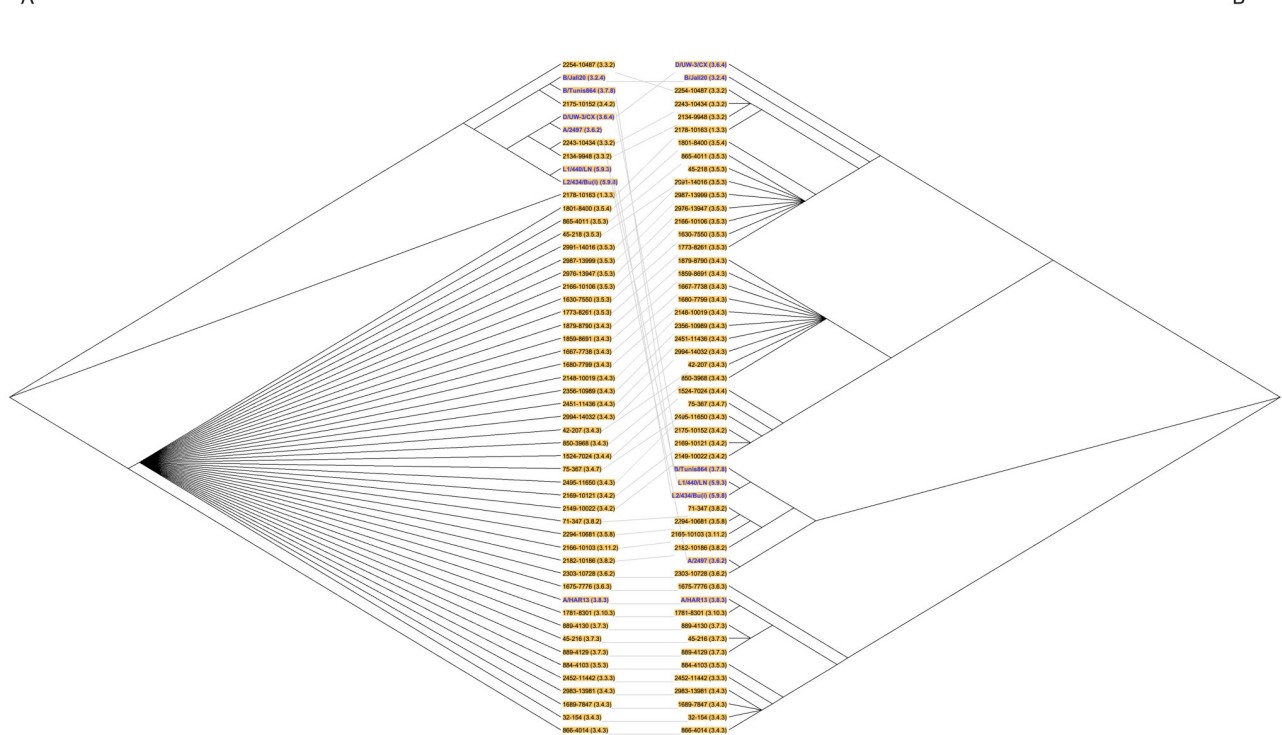

**Fig 5. Tanglegram showing the relation of the sequences derived from whole-genome sequencing (A) and Sanger sequencing (B).** Neighbor-Joining tree construction encompasses three variable number tandem repeat (VNTR) regions and associated flanking regions resulted from whole-genome sequencing (WGS) (A) and Sanger sequencing (B) of 45 *Chlamydia trachomatis* (Ct) positive samples collected in Ethiopia and seven Ct reference strains. All reference strains are highlighted in blue.

guideline value of DP = 0.95 [16]. MLVA-*omp*A detected a reduction in Ct diversity with additional years of "AFE" interventions which may mean that while the prevalence of infection remained high in this part of Amhara, the reduction in pathogen population diversity serves as an indicator of programmatic impact, possibly indicating the start of a decline in overall infection prevalence.

Region-wide efforts have been made over time to monitor trachoma and its risk factors within the Amhara region. These surveys have demonstrated that while WASH indicators have increased over time, efforts are still needed to achieve high levels of water and latrine availability and facial cleanliness regionwide [29,30]. It has been further demonstrated that in some areas of Amhara, TF and Ct prevalence remain stubbornly high despite many years of antibiotic MDA and F and E interventions [28–30,33]. A reduction in Ct diversity post-intervention as observed in this study, however, may be a precursor of sustained reduction in infection prevalence and reduce the chance of recrudescence, as it is likely that higher diversity affords the community of Ct variants more fitness in evasion [51]. A study in 2008 using samples from the Gambia investigated Ct diversity using *omp*A sequencing and suggested that diversity within trachoma Ct variants is lower in hypoendemic trachoma regions and showed that Ct diversity was lower post-MDA [11]. In a study based in the Gurage zone of Ethiopia [52], *omp*A diversity was not shown to be reduced post-MDA, however the discriminatory power of *omp*A alone is insufficient such that the diversity of the Ct community was not revealed. A 2019 study demonstrated mixed range of Ct diversity after one round of MDA,

**Table 5. Association analyses results comparing the frequency category of the variant with the absence or presence of TI or TF, the village level prevalence of TI or TF, and Ct load.** All models were adjusted for age and sex, with the "high" frequency group set as the reference level for all analyses. TI = Trachomatous inflammation-intense, TF = Trachomatous inflammation-follicular.

| Dependent variable | Variant frequency category | P value |
|---|---|---|
| Individual level TF and/or TI | High | Reference |
| | Medium | 0.20 |
| | Low | 0.12 |
| Village level TI | High | Reference |
| | Medium | 0.90 |
| | Low | 0.91 |
| Village level TF | High | Reference |
| | Medium | 0.66 |
| | Low | 0.64 |
| District level TI | High | Reference |
| | Medium | 0.43 |
| | Low | 0.58 |
| District level TF | High | Reference |
| | Medium | 0.92 |
| | Low | 0.33 |
| Individual Ct load | High | Reference |
| | Medium | 0.48 |
| | Low | 0.37 |

with a reduction in diversity in Senegal but not The Gambia [53]; the MLST DP used in this study was not as high as MLVA-*omp*A, and was well below the recommended 0.95 threshold. In our study MDA and other interventions appeared to have reduced Ct diversity, lending weight to the hypothesis that an observed reduction of Ct diversity could be a marker of treatment impact that precedes a reduction in overall prevalence. Further studies using samples collected after additional years of interventions in Amhara would be useful to continue to test this hypothesis.

Application of MLVA by Pedersen et al. [26] found DP = 0.94, which is consistent with other MLVA-*omp*A studies on genital serovars [54,55]. For LGV serovars, analyses testing the stability of VNTRs suggested MLVA to be a suitable typing method [36], however further testing by the same group on a larger set of LGV clinical samples suggested that MLVA-*omp*A could not reach the recommended 0.95 cut-off [15]. Other typing systems such as a combination of MLST and MLVA [22] and alternate MLVA targets [56] have been shown to have a DP of between 0.6–0.99. The typing system with the highest DP of 0.99, MLST+MLVA, requires 8 individual loci to be sequenced, double the number of this study, for a relatively marginal gain in DP. In addition, it is possible that when the mutation rate of a population is too high, a high level of DP provides too much resolution, and epidemiological links may be obscured [22]. In this study, the higher DP highlighted the difference between the diversity indices of the two timepoints in South Gondar. The *omp*A serovar type suggests a loss of diversity after extra rounds of MDA, with serovar A dominating in the second time point relative to the first, and the enhanced resolution provided by MLVA-*omp*A typing confirmed that the number of variants is significantly lower after extra rounds of MDA. This would suggest that MDA has driven purifying selection, however whether or not these STs persist in the population requires further monitoring. Surveys continue to monitor trachoma in Amhara, and it is recommended

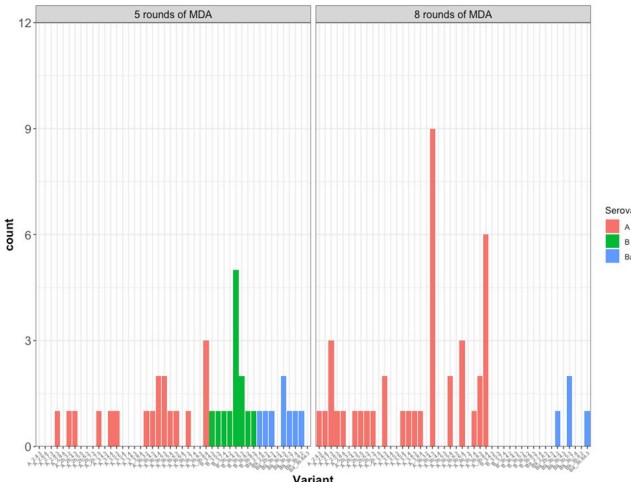

**Fig 6. Frequency plots for each *Chlamydia trachomatis* variant found in the South Gondar zone, Ethiopia, 2011–2017.** The data has been split into two time points- the first, after five rounds of MDA (n = 43, total number of variants = 30), and the second after an additional three to five rounds of MDA (n = 44, total number of variants = 23). Bars are coloured by each respective serovar. MDA = mass drug administration with azithromycin.

that these newly collected samples are MLVA-*omp*A typed to further understand the effect of MDA on diversity and the decline in infection prevalence.

Studies have shown that particular Ct variants have greater virulence [57,58], with identification of variants usually based on WGS. The relatively simpler and less expensive method of

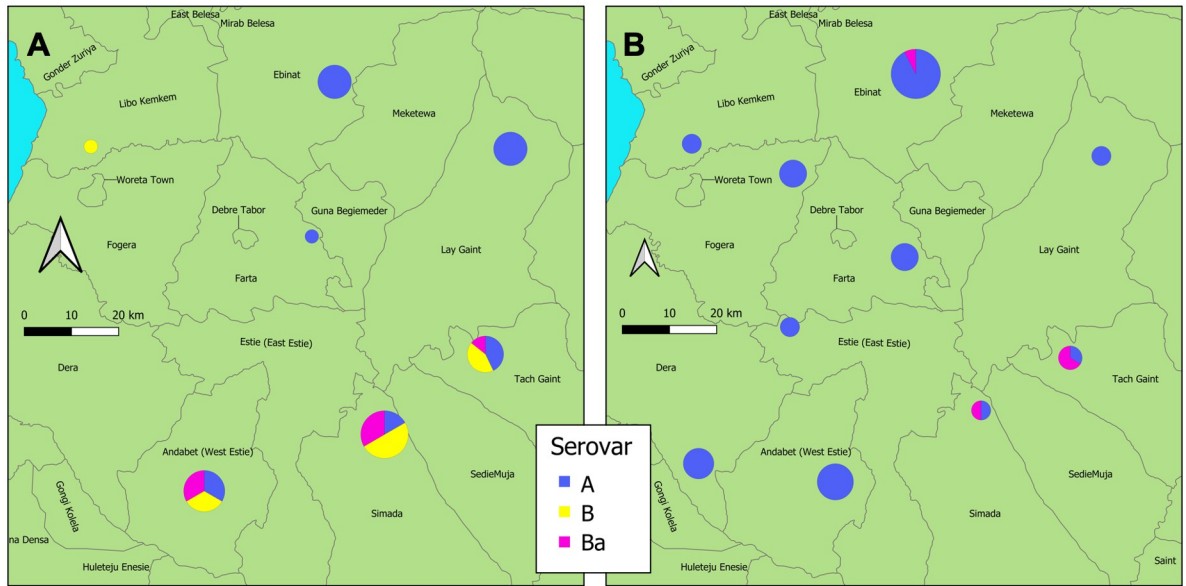

**Fig 7. Map of the South Gondar zone samples only (Amhara, Ethiopia), showing the range of *Chlamydia trachomatis* serovars identified from conjunctival swabs collected between 2011–2014 after five rounds of intervention (A) and eight to ten rounds of MDA (B) collected between 2015–2017.** The size of the pie chart is proportional to how many samples are represented, with larger pie charts indicating more samples. Maps were sourced from the Trachoma Atlas (https://atlas.trachomadata.org/) which uses Open Street Maps (Mapbox).

MLVA-*omp*A can provide sufficient resolution, enabling the study of intervention effects at the population level. We did not find significant associations between high, medium or low frequency STs and the clinical signs of trachoma or Ct load at village or district level. It is possible that due to the restricted number of samples included, there was insufficient statistical power to identify a relationship, however it is more likely that the regular rounds of MDA this population underwent has limited the likelihood of a single or common variant to become established and cause a significant increase in clinical signs of active trachoma. MSTs support this proposition as they demonstrate a homogenous distribution of variants within zones and across time (i.e. a lack of evidence for clustering of STs). While this study did not observe any association between STs and trachoma prevalence intensity, using this method in partnership with trachoma prevalence surveys would enable the detection of highly transmissible Ct strains if they were present, and enhanced interventions such as additional or targeted rounds of MDA along with improvements in facial cleanliness and environmental conditions could be deployed [59].

There are limitations to the MLVA-*omp*A typing method and our study. Firstly, mixed infections are not able to be identified since sequencing of a mixed product cannot be unequivocally interpreted, as mixed genotypes create mixed sequences [26]. Another possible limitation is the potential bias introduced by choosing samples with a higher load for sequencing, as this could affect the range of STs observed if some STs are associated with lower loads of infection. A final limitation was that some samples were not able to be assigned for CT1291 and CT1299, due to the length of the repeat cytosine (C)-region. DNA polymerase slippage is a well-known phenomenon where *taq* DNA polymerase disassociates from the template and reanneals at a matching base further along the strand, altering the original length of the repeat region [60]. This is part of the reason VNTRs have such high variation, as this happens naturally during DNA replication. CT1335, lacks repeat Cs and had a maximum number of 11 repetitions of T nucleotides so was not susceptible to slippage. Whether or not this is due to the number of C repeats or whether C itself drives polymerase slippage is unknown. In other studies, however, repeat regions of 21 Cs have been recorded [61], suggesting a methodological or DNA template issue. Short-read WGS struggles with repeat regions, this is due to problems with alignment of short read sequences and the algorithms used in next generation sequencing (NGS)/WGS pipelines [62,63]. This is reflected in the CT1291 and CT1299 VNTR sequences extracted from our WGS sequences from the samples in this study. In addition, the maximum likelihood phylogenetic tree constructed through WGS-derived sequences support this statement, wherein a significant portion of the sequences were indistinguishable and clustered together, irrespective of their associated STs. We found a high proportion of non-assigned bases within these two VNTR regions, as well as inconsistent results with the Sanger sequencing VNTR results. One possible reason is mapping ambiguity; When aligning short reads to a reference genome, the presence of VNTRs can lead to mapping ambiguity. If the number of repeats differs between the sample and the reference genome, it may be challenging to accurately map the reads to the correct genomic location [64]. As Sanger sequencing is often used to confirm the presence of single nucleotide polymorphisms found using WGS, it is assumed that the Sanger derived VNTR sequences in this study are the likely the more accurate [65].

The data from this study has shown that MLVA-*omp*A is a suitable technique for identifying ocular Ct variants on a local scale. While there were some issues surrounding long repeat regions, the typeability for all 3 VNTRs was approximately 65%. While the typeability of *omp*A variants was above 80%, this was after a second round of nested PCR in ~33% of samples, which involved further manipulation, financial costs and increased the risk of contamination. NGS techniques are associated with different issues, such as the requirement for more expensive equipment, more complicated analysis pipelines and computing power. It is difficult to

estimate the exact cost of WGS as genome size influences the number of samples that can be run in parallel, and a bioinformatician is generally required to analyse the data, a substantial part of overall costs [66]. The cost of performing WGS by Pickering et al., [27] was estimated at £250 per sample without post sequencing analysis, whereas the average cost of Sanger sequencing is estimated at ~£20 per sample for all four targets required for MLVA-*omp*A. Sanger sequencing also requires more readily available technology than WGS (since PCR amplicons can be simply posted to service providers). Additionally Sanger sequencing of a single product generates less data that is simpler to interpret with minimal training required.

This dataset has provided a unique collection of samples in which to investigate the utility of the MLVA-*omp*A method in a trachoma endemic region. Amhara is a region experiencing persistently high levels of trachoma [30], and is one of few programs that has been regularly monitoring Ct infection with ocular swabbing and PCR. The addition of Ct monitoring to trachoma programs not only allows for the tracking of Ct prevalence, but also allows for a deeper understanding of variant diversity, and the efficacy of MDA and other interventions, and it is recommended that other programs experiencing persistent or recrudescent trachoma include ocular Ct testing. The application of this typing technique provides an accessible, affordable and functional tool for tracking the spread and diversity of Ct variants over time and space, which can provide important information on the efficacy of MDA, aiding trachoma elimination programmes in resource limited settings.

## Supporting information

**S1 Fig. A minimum spanning tree showing the sequence-types identified in the Amhara region of Ethiopia between 2011–2017.** Each MLVA-*omp*A sequence-type (n = 87) is coloured by serovar. Each circle represents a different sequence-type, with the size of circle being directly proportional to the number of individuals who had that variant. Trees were generated using Phyloviz 2.0.
(TIF)

**S2 Fig. A minimum spanning tree showing the sequence-types identified in the Amhara region of Ethiopia between 2011–2017.** Each MLVA-*omp*A sequence-type (n = 87) is coloured by number of rounds of MDA, either 5 rounds (red) or 8–10 rounds (blue). Each circle represents a different sequence-type, with the size of circle being directly proportional to the number of individuals who had that variant. Trees were generated using Phyloviz 2.0.
(TIF)

**S3 Fig. A minimum spanning tree showing the sequence-types identified in the Amhara region of Ethiopia between 2011–2017.** Each MLVA-*omp*A sequence-type (n = 87) is coloured by zone. Each circle represents a different sequence-type, with the size of circle being directly proportional to the number of individuals who had that variant. Trees were generated using Phyloviz 2.0.
(TIF)

**S1 Table. The total number of Ethiopian conjunctival samples unable to be sequenced using Sanger sequencing for any of the three VNTRs and/or *omp*A for any reason (denoted as an "NA"), split by year of collection.** VNTR: Variable number tandem repeats.
(DOCX)

**S2 Table. WGS results from 99 samples vs Sanger sequencing results for the three VNTRs CT1291, CT1299 and CT1335.** WGS: Whole genome sequencing. VNTR: Variable number

tandem repeat.
(DOCX)

**S3 Table. The serovar and Pedersen sequence-type associated with each sample ID used to generate the phylogenetic trees (n = 45).**
(DOCX)

**S4 Table. The MLVA-ompA metadata containing individual level data on age group, gender, TF and TI, rounds of MDA, each VNTR type, the serovar and the associated typing code for each individual included in the study.**
(XLSX)

## Acknowledgments

Many thanks to the late Dr Julius Schachter and Jeanie Moncada for the kind gift of B/ Tunis864 original DNA. The authors thank Abbott for donation of the m2000 RealTime molecular diagnostics system and consumables.

## Author Contributions

**Conceptualization:** Ambahun Chernet, Eshetu Sata, Zerihun Tadesse, E. Kelly Callahan, Scott D. Nash, Martin J. Holland.

**Data curation:** Anna J. Harte, Harry Pickering, Joanna Houghton, Ambahun Chernet, Scott D. Nash.

**Formal analysis:** Anna J. Harte, Ehsan Ghasemian, Harry Pickering.

**Funding acquisition:** Zerihun Tadesse, E. Kelly Callahan.

**Investigation:** Anna J. Harte, Ambahun Chernet, Scott D. Nash.

**Methodology:** Anna J. Harte, Joanna Houghton, Ambahun Chernet, Scott D. Nash, Martin J. Holland.

**Project administration:** Anna J. Harte, Scott D. Nash, Martin J. Holland.

**Resources:** Eshetu Sata, Gizachew Yismaw, Taye Zeru, Zerihun Tadesse, E. Kelly Callahan.

**Supervision:** Joanna Houghton, Eshetu Sata, Gizachew Yismaw, Taye Zeru, Zerihun Tadesse, E. Kelly Callahan, Scott D. Nash, Martin J. Holland.

**Visualization:** Anna J. Harte.

**Writing – original draft:** Anna J. Harte, Scott D. Nash, Martin J. Holland.

**Writing – review & editing:** Anna J. Harte, Ehsan Ghasemian, Harry Pickering, Joanna Houghton, Ambahun Chernet, Eshetu Sata, Gizachew Yismaw, Taye Zeru, Zerihun Tadesse, E. Kelly Callahan, Scott D. Nash, Martin J. Holland.

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
