## [Decision Letter · Decision Letter 0]

7 Mar 2024

Dear Dr. Harte,

Thank you very much for submitting your manuscript "Unravelling Chlamydia trachomatis Diversity in Amhara, Ethiopia: MLVA-ompA Sequencing as a Molecular Typing Tool for Trachoma" for consideration at PLOS Neglected Tropical Diseases. As with all papers reviewed by the journal, your manuscript was reviewed by members of the editorial board and by several independent reviewers. The reviewers appreciated the attention to an important topic. Based on the reviews, we are likely to accept this manuscript for publication, providing that you modify the manuscript according to the review recommendations. 

Sincerely,

Jeremiah M. Ngondi, MB.ChB, MPhil, MFPH, Ph.D

Academic Editor

Ana LTO Nascimento

Section Editor

Reviewer's Responses to Questions

**Key Review Criteria Required for Acceptance?**

**Methods**

-Are the objectives of the study clearly articulated with a clear testable hypothesis stated?

-Is the study design appropriate to address the stated objectives?

-Is the population clearly described and appropriate for the hypothesis being tested?

-Is the sample size sufficient to ensure adequate power to address the hypothesis being tested?

-Were correct statistical analysis used to support conclusions?

-Are there concerns about ethical or regulatory requirements being met?

Reviewer #1: The methods are excellent, except I believe that if they are trying to understand why the rates of trachoma had not fallen they should have accessed changes in facial cleanliness rates to have a measure of transmission potential.

Reviewer #2: In the manuscript, the authors investigated the genetic diversity of Chlamydia trachomatis in Amhara, Ethiopia, and compared this diversity at two timepoints: during 5 rounds and 8-10 rounds of mass drug administration (MDA) using VNTR combined with ompA genotyping. The methodology employed in the study appears to be appropriate for the research objectives and the authors utilized sufficient statistical methods required for analyzing genetic data.

Reviewer #3: All appropriate. Stats could be expanded as mentioned below

**Results**

-Does the analysis presented match the analysis plan?

-Are the results clearly and completely presented?

-Are the figures (Tables, Images) of sufficient quality for clarity?

Reviewer #1: The results are excellent

Reviewer #2: The analysis presented in the study aligns with the outlined analysis plan and the results are clearly and thoroughly presented. However, I have a few suggestions to further enhance the clarity and impact of the results:

1- For comparison with MLVA-ompA, it would be beneficial to include a minimum spanning tree (MST) of ompA and VNTR sequences overlaid with serovar, zone, and date of sample collection in the supplementary data.

2- I recommend the authors provide a Tanglegram comparing tree topology differences between whole-genome sequencing (WGS) extracted and Sanger sequencing VNTR, instead of presenting two phylogeny trees in figures 5 and 6.

3- It would enhance the analysis if the authors included a Tanglegram comparing MLVA-ompA sequencing versus ompA sequencing.

Reviewer #3: Yes. could be trimmed, but not necessary for online journal?

**Conclusions**

-Are the conclusions supported by the data presented?

-Are the limitations of analysis clearly described?

-Do the authors discuss how these data can be helpful to advance our understanding of the topic under study?

-Is public health relevance addressed?

Reviewer #1: The conclusion that their sequencing is much cheaper that WGS is sound. 

However, in the abstract and discussion there are several areas where the influence of the lack of facial cleanliness on the potential for transmission has been omitted. Lines 16-17, 42-43 and in part 61-62 gloss over the importance of facial cleanliness as the long-term measure to reduce trachoma transmission and reinfection. If in lines 573-575 they wanted to find out why rates of trachoma had not fallen or had recrudesced they should be measuring facial cleanliness.

What is totally unclear is how their sequencing will be ‘aiding in the development of targeted interventions for trachoma control’ (Lines 37,53, and much of the discussion). This is especially so as they did not find any association between STs and trachoma prevalence of intensity, Line 518-519 and in the results.

Reviewer #2: The conclusions drawn in the study are well-supported by the data presented. The authors effectively outline the limitations of their analysis. Furthermore, the study addresses the public health relevance of its findings.

Reviewer #3: Could add in some limitations. a few listed below. could add in interpretation of decreased diversity. again, as below

**Editorial and Data Presentation Modifications?**

Reviewer #1: I don’t know if the authors have facial cleanliness data but if they do they could improve this paper significantly.

Reviewer #2: 1- Please review Table 4A for any typos. I am uncertain if the last ST should be listed as 4* or 14*.

2- I couldn't find the sequences in accession number provided. Please check the accession number for typo error.

Reviewer #3: (No Response)

**Summary and General Comments**

Reviewer #1: (No Response)

Reviewer #2: (No Response)

Reviewer #3: This thorough analysis of chlamydia variants in a trachoma-endemic area covers a number of issues: phylogenetics, minimum spanning trees, spatiotemporal clustering, etc. While these are well worth reporting, a particular point of interest is the apparent decrease in diversity with MDA, as it may have clinical ramifications. If diversity helps evade the human host response, then lower diversity post MDA may prevent chlamydia from returning to pre-treatment levels. I believe that your finding of decreased diversity is important. The authors could consider expanding on their important finding of a decrease. Either in a few further analyses, or in the discussion. Or both! Either would be of interest though. 

1) Community-level diversity across different bacterial genera and species has decreased in RCTs of azithromycin MDA. Earlier studies suggested that diversity within trachoma variants of chlamydia might be lower with lower prevalence across regions, and hypothesized that diversity might be lower post-MDA. But this was not found in at least one trachoma RCT (Chin et al, AJE, 2018). This highlights the importance of your results.

2) The high discriminatory power (which I think is synonymous with Simpson’s diversity or 1-Simpson’s index if I understand correctly) and your Figure 7 imply the vast majority of variants were unique. The most information about diversity can sometimes be obtained by operational taxonomic units that are neither too broad (can’t distinguish differences) nor too narrow (find every sample is unique, as found in Guinea Bissau). You have the opportunity to try several operational taxonomic units of different discriminatory power. I may have missed, but did you compare pre- and post-MDA with each of your diversity measures?

3) Fine to display as 1-Simpson’s index, with interpretation as your discriminatory power (again, if I understand correctly). Another natural unit is as “effective number”, which would be the reciprocal of Simpson’s index (L-2 Renyi entropy or Hill number, reviewed by Lou Jost in 2006, 2008, 2010). Also, could consider a sensitivity analysis with Shannon’s (consider expressing as e^Shannon’s, which would be “effective number”). None of these are necessary. And I think you provide the data for interested readers to check themselves? Or it could be expanded on in subsequent publications by your group.

Minor comments:

4) Abstract. “identification of CT genotypes is a necessity”. Consider “could be useful”. Trachoma may well be eliminated before chlamydia diversity is understood.

5) Selecting samples with higher CT load might induce some bias, line 146-7. Could be mentioned in limitations.

PLOS authors have the option to publish the peer review history of their article (what does this mean?). If published, this will include your full peer review and any attached files.

Reviewer #1: No

Reviewer #2: No

Reviewer #3: No

Figure Files:

Data Requirements:

Reproducibility:

References

---

## [Editor Report · Decision Letter 1]

8 Apr 2024

Dear  Dr. Harte,

We are pleased to inform you that your manuscript 'Unravelling Chlamydia trachomatis Diversity in Amhara, Ethiopia: MLVA-ompA Sequencing as a Molecular Typing Tool for Trachoma' has been provisionally accepted for publication in PLOS Neglected Tropical Diseases.

Best regards,

Jeremiah M. Ngondi, MB.ChB, MPhil, MFPH, Ph.D

Academic Editor

Ana LTO Nascimento

Section Editor

---

## [Editor Report · Acceptance letter]

17 Apr 2024

Dear Dr Harte,

We are delighted to inform you that your manuscript, "Unravelling *Chlamydia trachomatis* Diversity in Amhara, Ethiopia: MLVA-ompA Sequencing as a Molecular Typing Tool for Trachoma," has been formally accepted for publication in PLOS Neglected Tropical Diseases.

Best regards,

Shaden Kamhawi

co-Editor-in-Chief

Paul Brindley

co-Editor-in-Chief
